# Nonalcoholic Fatty Liver Disease and Altered Neuropsychological Functions in Patients with Subcortical Vascular Dementia

**DOI:** 10.3390/jpm12071106

**Published:** 2022-07-05

**Authors:** Rita Moretti, Mauro Giuffré, Lory Saveria Crocè, Silvia Gazzin, Claudio Tiribelli

**Affiliations:** 1Department of Medical, Surgical, and Health Sciences, University of Trieste, 34149 Trieste, Italy; moretti@units.it (R.M.); lcroce@units.it (L.S.C.); 2Italian Liver Foundation, Centro Studi Fegato, 34149 Trieste, Italy; silvia.gazzin@fegato.it (S.G.); ctliver@fegato.it (C.T.)

**Keywords:** liver steatosis, NAFLD, small vessel disease, subcortical vascular dementia, vascular risk factors, behavior diseases

## Abstract

NAFLD is the most common cause of abnormality in liver function tests. NAFLD is considered a potential cardiovascular risk factor and is linked to cardiovascular risk factors such as obesity, hypertension, type 2 diabetes, and dyslipidemia. Few previous studies have investigated whether NAFLD could be independently associated with cognitive impairment. The current study aims to find a possible role of NAFLD in the development of subcortical vascular dementia (sVaD). We considered NAFLD as a possible independent vascular risk factor or, considering its metabolic role, associated with other commonly accepted sVaD risk factors, i.e., lack of folate, vitamin B12, and vitamin D-OH25, and increased levels of homocysteine. We studied 319 patients diagnosed with sVaD. All patients underwent an abdominal ultrasound examination to classify steatosis into four levels (1—none up to 4—severe). sVaD patients were divided into two groups according to the presence or absence of NAFLD. Our results demonstrated a strong correlation between NAFLD and sVaD. Patients with the two comorbidities had worse neuropsychological outcomes and a worse metabolic profile. We also found a robust relationship between NAFLD and severe vitamin B12, folate, vitamin D hypovitaminosis, and higher hyperhomocysteinemia levels. This way, it is evident that NAFLD contributes to a more severe metabolic pathway. However, the strong relationship with the three parameters (B12, folate and vitamin D, and homocysteinemia) suggests that NAFLD can contribute to a proinflammatory condition.

## 1. Introduction

Nonalcoholic Fatty Liver Disease (NAFLD) is characterized by excessive hepatic fat accumulation. It is defined by steatosis in >5% of hepatocytes in the absence of other concurrent etiologies [1,2,3,4,5]. It encompasses a broad spectrum of conditions, which goes from the sole condition of liver steatosis (NAFLD) to a more complex—and prognostically worse—condition of steatosis associated with inflammation and hepatocyte ballooning, in which fibrosis may or may not be present, referred to as Nonalcoholic SteatoHepatitis (NASH) [6], which covers a broad spectrum of disease severity. It also carries a higher risk of the development of cirrhosis, liver failure, or hepatocellular carcinoma (HCC).

NAFLD is the most common cause of abnormality in liver function tests, and due to its high prevalence, it has become one of the most common indicators of a liver transplant. It is likely a relevant cause of the so-called ‘cryptogenic’ cirrhosis. Currently, nearly 40% of previously diagnosed cryptogenic cirrhosis accounts for NAFLD [7]. NAFLD is a potential cardiovascular risk factor related to obesity, hypertension, type 2 diabetes, and dyslipidemia [8,9,10,11].

Won Seo et al. [12] found out that NAFLD was independently associated with lower cognitive performance, in terms of age-related cognitive decline, independently of cardiovascular risk factors. Further, in our previous work [13], we reported that NAFLD could be associated with an evident alteration of rapidity and precision of executive functions and a higher degree of behavioral disturbances and anxiety in a group of patients referred to our neurological department and not affected by dementia.

The cerebrovascular disease significantly contributes to cognitive decline in the elderly [14]. The most common cause of subcortical vascular dementia (sVaD) is cerebral small vessel disease (SVD), which typically manifests as white matter lesions or lacunes on brain imaging due to distress of small perforating arteries <50 μm, which supply deep brain structures [14]. The primary aim of the current study is to find a possible role of NAFLD in the development of sVaD as an independent vascular risk factor or by association with other commonly widely accepted sVaD risk factors, such as folate, vitamin B12, and vitamin D-OH25 deficiency, as well as increased levels of homocysteine [15,16,17,18,19]. The secondary aim is to define the association between NAFLD and the degree of dementia in a recently diagnosed sVaD population.

## 2. Subjects Characteristics

The current study included 319 adult patients diagnosed with sVaD and referred to the Neurological Clinic of Cattinara University Hospital (Trieste) from 1 June 2014 to 1 June 2019.

The diagnosis of subcortical vascular dementia was made according to the NINDS AIREN criteria [20,21,22]. sVaD was diagnosed if CT/MRI scans showed moderate to severe ischemic white matter changes and at least one lacunar infarction [23,24]. All patients showed severe white matter hyperintensities on MRI, localized around the lateral ventricles or within the deep white matter [25,26]. Brain CT scans or MRI images were available for all the patients; a neurologist (RM) revised all the imaging, employing the Blennow scale for CT scans [27,28] and the Fazekas scale for MRI imaging [24] according to the most recent findings [29,30].

Patients were excluded from the study if they showed signs of normal pressure hydrocephalus (by history, clinical examination, and by the absence of axial FLAIR MRI scan showing a significant ventriculomegaly with increased Evans Index, the ratio of a maximum width of the frontal horns of the lateral ventricles and maximal internal diameter of the skull at the same level on axial CT or MRI images, absence of the T1-weighted coronal gadolinium-enhanced MRI scan showing reduced callosal angle, and absolute absence of the Axial FLAIR MRI scan revealing enlarged lateral ventricles with bright signal in the surrounding white matter, suggestive of transependymal edema [31,32]); previous brain tumors and/or the previous diagnosis of a major cerebrovascular event; white matter lesions, caused by different specific etiologies, such as multiple sclerosis, collagen vascular disease, and genetic forms of vascular dementia; or with alcoholism.

Study subjects underwent a detailed history, physical examination, neurological examination, laboratory tests, and abdominal ultrasound and completed the test with a series of neuropsychological examinations. Body mass index (BMI) was calculated for all participants.

Standard questionnaires were conducted with all participants and included alcohol consumption, smoking, drug use, and physical activity. We obtained the average daily estimate for alcohol consumption as less than 20 g/day for women and less than 30 g/day for men [33,34,35].

Liver enzymes, alanine aminotransferase (ALT), aspartate aminotransferase (AST), as well as a second-generation enzyme immuno-assay (Abbott Laboratories, Chicago, IL, USA) were employed to detect the presence of hepatitis C antibodies. At the same time, a solid-phase competitive immuno-assay (Abbott Laboratories, Milano, Italy) was used to detect antibodies to the hepatitis B core antigen.

We evaluated fasting venous blood sample, which was collected, centrifuged immediately, and stored at −80 °C for further laboratory analysis. Clinical measurements included the following: total serum cholesterol, triglycerides, high-density lipoprotein (HDL), and low-density lipoprotein (LDL), which were calculated by Friedwald’s formula [36]. Blood folate, vitamin B12, and homocysteine serum were also tested as specific measures of vascular risk factors [37,38]. We have considered, following average literature values: Folate (3.8–26 ng/mL), Vitamin B12 (201–870 pg/mL), Homocysteine (3–15 mcmol/L) [39,40,41].

Diabetes mellitus was defined as venous plasma glucose concentrations of >120 mg/dL after an overnight fast. Glycated hemoglobin (Hb1Ac) results were aligned to the assay applied by the Diabetes Control and Complication Trial (DCCT), expressed as a percentage (normal range: 4–5.6%; 5.5–6.5% indicates a high risk of developing diabetes; a value superior to 6.6% indicates diabetes) [42,43]. Serum levels of 25 (OH)D vitamins were measured using enzyme immuno-assay kits (DIA Source ImmunoAssay, SA Belgium). We accepted the National Osteoporosis Society parameters [19] (Vitamin D-OH25 deficiency ≤ 12 ng/mL; Vitamin D-OH25 insufficiency 12–20 ng/mL; Vitamin D-OH25 sufficiency ≥ 20 ng/mL).

Besides, all the recruited participants underwent a complete abdominal ultrasound with a TOSHIBA SSA-907 machine employing a 5-Mhz probe. The grading of liver steatosis was obtained by evaluating liver brightness, liver-to-kidney contrast, the appearance of intrahepatic vessels, and diaphragm definition. Steatosis was graded as follows: absent (score = 0), when the liver echotexture is normal; mild (score = 1), when there is a slight increase in liver echogenicity with standard visualization of the diaphragm and liver vessels; moderate (score = 2), in case of a moderate increase in liver echogenicity with the slightly impaired appearance of diaphragm and liver vessels; severe (score = 3), in case of markedly increased liver echogenicity with poor/no visualization of liver vessels or diaphragm.

All the patients underwent a simple sequence of neuropsychological tests:Frontal Assessment Battery (FAB; score: 0–18; 18 = normal) [44,45];Hamilton Anxiety Rating Scale (HAM-A) (score: 0–56; a total score comprised 0–17, estimated mild anxiety; 18–24, mild to moderate anxiety; 25–30, severe anxiety) [46];Apathy Evaluation Score (AES-C) (clinical examination; score: 18–72; higher scores reflect more apathy) [47];Global behavioral symptoms, assessed by the Neuropsychiatric Inventory, NPI [48]; NPI was registered either as a total score, or as specific scores, with specific mention of depression, hallucinations, and delusions (frequency and intensity of symptoms, with the correct score of 4 × 3, considering a maximum score of 12).Quality of life in dementia scale (QUALID) [49]. The proxy rating scale consists of 11 items rated on a five-point scale. The items are rated by frequency of occurrence, comprising both positive and negative dimensions of concrete and observable mood and performance. Scores are summed to range from 11 to 55. A lower score indicates a higher quality of life.

The present study was conducted following the Declaration of Helsinki and under the Ethics Guidelines (Point 4 of the CEUR Declaration) of the Committee of the University–Hospital of Trieste. Written informed consent was obtained from all the participants.

## 3. Statistical Analysis

The Shapiro–Wilk test was performed to verify the normal distribution of variables. According to the test results, continuous variables were reported as medians (Quartile 1; Quartile 3), while discrete variables were reported as the number and proportion of subjects with the characteristic of interest. Before categorization, intergroup (absent vs. mild NAFLD) differences in distribution were investigated, resulting in the absence of statistically significant differences between these two groups. Therefore, patients were divided into two groups: group A, which included patients with moderate-to-severe NAFLD, and group B, which included patients with non-to-mild NAFLD. Intergroup differences were investigated using the Mann–Whitney U test, Wilcoxon Rank-Sum test for continuous variables, and Pearson’s Chi-Square test for discrete variables. Correlations between continuous variables and NAFLD severity were studied with a point-biserial correlation coefficient. Patients were then categorized, despite their NAFLD status, into subgroups according to the median values of each neuropsychological test (higher than median = 1; lower than the median value = 0). Variables were modeled univariately using binary logistic regression to study the Odds ratios and, therefore, investigate factors associated with worse neuropsychological test results. Variables that resulted in statistically significant data univariately were further studied multivariately, and the best model was chosen for the lowest Bayesian information criterion (BIC) [50]. For all analyses, two-sided statistical significance was defined as *p* < 0.05. Data were analyzed using SPSS (Statistical Package for Social Science) version 26.0 (IBM SPSS Statistics for MAC OS. Armonk, NY, USA: IBM Corp.).

## 4. Results

We recruited 319 consecutive patients. Thirty-four of them abandoned the study or were excluded for lack of compliance. A total of 285 patients were affected by sVaD.

A total of 285 patients were affected by sVaD; patients were then divided into two subgroups according to their NAFLD status: Group A (n = 114) included patients with moderate (n = 106, 93%) and severe (n = 8, 7%) NAFLD, while Group B (n = 171) included patients without (n = 148, 86.5%) or with mild (n = 23, 13.5%) NAFLD.

As reported in Table 1, patients were predominantly male (n = 148, 51.9%), with a median age of 76 (75;76). Statistically significant differences were found between patients with and without NAFLD in median education level, HbA1c, LDL, Vitamin D, folate, homocysteine, ALT, AST, and BMI.

As reported in Table 2, in terms of neuropsychological tests, patients with and without NAFLD showed statistically significant, different results in the median FAB test (*p* < 0.001), QUALID test (*p* < 0.001), AES-C test (*p* < 0.001), HAM-A test (*p* < 0.001), and NPI test (*p* < 0.001).

As reported in Table 3, a strong positive correlation was found between NAFLD severity and BMI, ALT, AST, QUALID test, AES-C test, HAM-A test, and NPI test. At the same time, a moderate positive correlation was found between NAFLD severity and HbA1C, homocysteine, and LDL. On the contrary, a strong negative correlation was found between NAFLD severity and vitamin D levels. A moderate negative correlation was found between NAFLD severity and folate and the FAB test. In contrast, a weak correlation was found between education level and a very weak correlation with Vitamin B12.

## 5. Predictors of Worse Neuropsychological Test Results

Despite their NAFLD status, patients were stratified according to median neuropsychological test results. Each studied variable was investigated univariately and then modeled multivariately as a predictor of worse neuropsychological test outcomes. The results of binary logistic regression are reported in Table 4 for the FAB test, Table 5 for QUALID Test, Table 6 for the AES-C test, Table 7 for the HAM-A test, and Table 8 for the NPI test.

## 6. Discussion

In this cross-sectional study comprising 285 sVaD patients, patients with moderate-to-severe NAFLD (Group A) had worse metabolic profiles than non-to-mild NAFLD (Group B). NAFLD is associated with metabolic aspects and causes a lack of vitamin B12, folate, and vitamin D levels, with a higher homocysteine level and increased common vascular risk factors, i.e., body mass index, LDL cholesterol, and elevation of glycated hemoglobin. These results align with previous findings [13,51,52,53,54,55,56,57,58,59,60] but additionally show a positive correlation between NAFLD severity and higher apathy, anxiety, behavioral worsening, and decreased quality of life; a negative correlation with executive dysfunctions; and a strong correlation with metabolic parameters in patients with subcortical vascular dementia.

NAFLD is strictly associated with obesity and higher glycated levels [61,62], with lower levels of folate and vitamin B12 [63,64], and a reduction in folate level [65]. More recently, NAFLD has been associated with higher levels of homocysteine [66] and most probably hypovitaminosis D [13,66].

Hyperhomocysteinemia and reduced levels of folate, B12, and D vitamins have been related to sVaD’s worse clinical outcomes [67,68,69,70,71,72,73,74,75,76,77,78,79,80]. Transgenic and wild-type mice suffering from Alzheimer’s disease (AD) were exposed to a high-fat diet for up to twelve months [81]. Both developed NALFD and systemic signs, plus cerebral signs of inflammation; however, only AD transgenic mice were observed to have an accelerated Abeta plaque deposition. Removal of a high-fat diet after two months decreased Abeta plaque load in transgenic mice and reversed signs of systemic and cerebral inflammation in both groups [82]. A progressive neural and glial death occurred, related to increased neurofibrillary tangles depositions and cerebral amyloid angiopathy all around the cerebral vessels [81]. This relationship is significant to consider NAFLD as a risk factor for white matter alteration and a causative factor of sVaD. It could be considered a long-term predictor of overall dementia in patients with other physical frailties [80,82,83,84,85,86,87,88,89,90,91,92,93,94,95,96,97,98,99].

Our study testifies a strong association between NAFLD and a general worsening of the neuropsychological signs and symptoms in patients meeting the criteria for sVaD.

Our results seem contradicted by another retrospective matched cohort study of 656 NAFLD patients during a mean follow-up of 19 years [100]. A total 3.3% of the NAFLD patients and 4.9% of the controls developed dementia, suggesting that the link between the two pathologies is poorly defined [100].

By contrast, a recent nationwide study found that NAFLD is associated with an increased risk of dementia [101]. The association was more pronounced among females and nonobese NAFLD subjects [89]. Despite the many limitations of the study, the results suggest a possible link between NAFLD development in an abnormal fat metabolism and fat dysregulation, coinciding with NAFLD and dementia [101]. These results have been recently confirmed in animal models [102].

Weight gain, obesity, and NAFLD are increasing worldwide [103], and the brain consequences should be studied. At the same time, vascular dementia is becoming one of the most studied neurological conditions in aging people [88]. NAFLD could be prevented, which might interfere with further neurological complications.

Our study has several limitations. It is a single-center cross-sectional study with a limited number of patients, without histological assessment. On the other hand, to the best of our knowledge, this is the first study exploring the possible relationship between NAFLD and sVaD [1,6,11,104].

More multicenter and dedicated studies would be necessary to understand this exciting and unknown perspective fully.

## Figures and Tables

**Table 1 jpm-12-01106-t001:** Clinical, biochemical, and Anthropometric characteristics of the enrolled population (*n* = 285). Continuous variables are reported by median and interquartile ranges (IQR). Patients are stratified by presence (*n* = 114) and absence (*n* = 171) of NAFLD.

Variable (Normal Values)	All Patients (n = 285)	Group A (NAFLD), (n = 114)	Group B (without NAFLD), (n = 171)	Significance
Age, years	76 (75; 76)	76 (75; 76)	76 (75; 77)	NS
Educational level, years	11 (10; 12)	9 (8; 12)	12 (11; 12)	*p* < 0.001
Gender M, n %	148 (51.9%)	59 (51.7%)	89 (52%)	NS
HbA1C, %	5.9 (5.7; 6.5)	6.7 (6.3; 6.9)	5.7 (5.4; 5.9)	*p* < 0.001
LDL, mg/dL	122.7 (112.3; 132.4)	132.9 (5.1)	112.4 (109.8; 121.4)	*p* < 0.001
Vitamin DOH25, mg/dL	12.3 (9.5; 17.8)	9.4 (9.2; 9.7)	16.5 (12.5; 17.9)	*p* < 0.001
Folate, ng/mL	2.5 (2.3; 3.8)	2.3 (2.1; 2.4)	3.1 (2.5; 4.2)	*p* < 0.001
Vitamin B12, pg/mL	145.4 (129.4; 154.3)	123.4 (122.4; 132.4)	154.3 (145.4; 157.6)	*p* < 0.001
homocysteine, µmol/L	19.2 (15.3; 23.4)	23.4 (22.3; 24.1)	15.4 (14.1; 18.95)	*p* < 0.001
ALT (IU/L)	33 (29; 43)	44 (43; 45)	29.8 (28.75; 32.2)	*p* < 0.001
AST (IU/L)	34 (32; 43)	44 (43; 46)	33.1 (29.4; 34.3)	*p* < 0.001
BMI	24 (23; 34)	34 (34; 35)	23 (22; 24)	*p* < 0.001

**Abbr.** HbA1C—glycate hemoglobin; LDL—low-density lipoprotein; ALT—alanine aminotransferase; AST—aspartate aminotransferase; BMI—body mass index.

**Table 2 jpm-12-01106-t002:** Neuropsychological test characteristics of the enrolled population (*n* = 285). Continuous variables are reported by median and interquartile ranges (IQR). Patients are stratified by presence (*n* = 114) and absence (*n* = 171) of NAFLD.

Characteristics	All Patients (n = 285)	Group A (NAFLD), (n = 114)	Group B (without NAFLD), (n = 171)	Significance
FAB test	12 (11; 13)	11 (10; 12)	13 (12; 13)	*p* < 0.001
QUALID Test	24 (15; 28)	31 (26; 33)	23 (14; 24)	*p* < 0.001
AES-C	24 (23; 37)	38 (36; 39)	23 (22; 24)	*p* < 0.001
HAM-A Test	22 (16; 27)	30 (26; 32)	18 (16; 21)	*p* < 0.001
NPI	24 (18; 36)	39 (27; 42)	21 (18; 24)	*p* < 0.001

**Abbr:** FAB—Frontal Assessment Battery; QUALID—Quality of life in dementia scale; AES-C—Apathy Evaluation Score; HAM-A—Hamilton Anxiety Rating Scale; NPI—Neuropsychiatric Inventory.

**Table 3 jpm-12-01106-t003:** Correlations between NAFLD severity and the measured variables investigated through point-biserial correlation test.

Variable	Strength (rbp)	Significance
Age, years	NS	NS
Education Level, years	−0.406	*p* < 0.001
BMI, kg/m^2^	0.914	*p* < 0.001
ALT, IU/L	0.823	*p* < 0.001
AST, IU/L	0.831	*p* < 0.001
HbA1C, %	0.669	*p* < 0.001
Vitamin D, mg/dL	−0.710	*p* < 0.001
Homocysteine, µmol/L	0.692	*p* < 0.001
Folate, ng/mL	−0.613	*p* < 0.001
Vitamin B12, pg/mL	−0.132	*p* < 0.001
LDL, mg/dL	0.648	*p* < 0.001
FAB test	−0.678	*p* < 0.001
QUALID Test	0.717	*p* < 0.001
AES-C	0.900	*p* < 0.001
HAM-A Test	0.800	*p* < 0.001
NPI	0.752	*p* < 0.001

**Table 4 jpm-12-01106-t004:** Factors associated with worse FAB test results in terms of Odds Ratio and its 95% C.I. Variables are initially studied univariately and then multivariately by choosing the model with the lowest BIC. Abbreviations not previously explained: NAFLD—nonalcoholic fatty liver disease.

FAB Test	Univariate Analysis		Multivariate Analysis	
Variables	Odds Ratio	Significance		
Age, years	0.84 (0.68–1.04)	*p* = 0.108		
Education Level, years	0.63 (0.54–0.735)	*p* < 0.001		
BMI, kg/m^2^	1.37 (1.29–1.57)	*p* < 0.001		
ALT, IU/L	1.29 (1.22–1.36)	*p* < 0.001		
AST, IU/L	1.35 (1.27–1.44)	*p* < 0.001		
Hb1Ac (%)	8.01 (4.66–13.19)	*p* < 0.001	3.14 (2.58–6.56)	*p* < 0.001
Vitamin D, mg/dL	0.56 (0.47–0.66)	*p* < 0.001	0.55 (0.40–0.60)	*p* < 0.001
Homocysteine µmol/L	1.45 (1.32–1.59)	*p* < 0.001		
Folate, ng/mL	0.05 (0.02–0.15)	*p* < 0.001		
Vitamin B12, pg/mL	0.93 (0.91–0.95)	*p* < 0.001		
LDL, mg/dL	1.09 (1.07–1.12)	*p* < 0.001		
NAFLD Severity	7.565 (7.55–10.78)	*p* < 0.001	6.92 (5.10–7.92)	*p* < 0.001

**Table 5 jpm-12-01106-t005:** Factors associated with worse QUALID test results in terms of Odds Ratio and its 95% C.I. Variables are initially studied univariately and then multivariately by choosing the model with the lowest BIC.

QUALID Test	Univariate Analysis		Multivariate Analysis	
Variables	Odds Ratio	Significance	Odds Ratio	Significance
Age, years	0.89 (0.73–1.01)	*p* = 0.287		
Education Level, years	0.65 (0.57–0.75)	*p* < 0.001		
BMI, kg/m^2^	1.54 (1.41–1.68)	*p* < 0.001		
ALT, IU/L	1.33 (1.25–1.41)	*p* < 0.001		
AST, IU/L	1.50 (1.37–1.64)	*p* < 0.001		
Hb1Ac (%)	7.66 (6.11–9.63)	*p* < 0.001	2.55 (1.02–6.4)	*p* = 0.03
Vitamin D, mg/dL	0.66 (0.59–0.73)	*p* < 0.001	0.50 (0.43–0.60)	*p* = 0.04
Homocysteine µmol/L	1.46 (1.34–1.58)	*p* < 0.001	1.41(1.20–1.30)	*p* = 0.02
Folate, ng/mL	0.18 (0.11–0.29)	*p* < 0.001		
Vitamin B12, pg/mL	0.91 (0.89–0.93)	*p* < 0.001		
LDL, mg/dL	1.10 (1.07–1.13)	*p* < 0.001		
NAFLD Severity	9.40 (6.18–14.30)	*p* < 0.001	5.39 (2.94–9.89)	*p* < 0.001

**Table 6 jpm-12-01106-t006:** Factors associated with worse AES-C test results in terms of Odds Ratio and its 95% C.I. Variables are initially studied univariately and then multivariately by choosing the model with the lowest BIC.

AES-C Test	Univariate Analysis		Multivariate Analysis	
Variables	Odds Ratio	Significance	Odds Ratio	Significance
Age, years	0.86 (0.70–1.05)	*p* = 0.141		
Education Level, years	0.66 (0.58–0.77)	*p* < 0.001		
BMI, kg/m^2^	1.83 (1.54–2.20)	*p* < 0.001		
ALT, IU/L	1.38 (1.28–1.48)	*p* < 0.001		
AST, IU/L	1.55 (1.39–1.72)	*p* < 0.001		
Hb1Ac (%)	8.15 (6.95–12–93)	*p* < 0.001	4.61 (1.48–14.4)	*p* = 0.008
Vitamin D, mg/dL	0.62 (0.56–0.70)	*p* < 0.001	0.70 (0.61–0.85)	*p* = 0.003
Homocysteine µmol/L	1.54 (1.41–1.70)	*p* < 0.001	1.207 (1.1–1.34)	*p* = 0.001
Folate, ng/mL	0.14 (0.08–0.23)	*p* < 0.001	0.23 (0.15–0.40)	*p* = 0.043
Vitamin B12, pg/mL	0.93 (0.91–0.94)	*p* < 0.001		
LDL, mg/dL	1.13 (1.10–1.68)	*p* < 0.001		
NAFLD Severity	10.98 (6.92–17.40)	*p* < 0.001	3.89 (2.01–7.5)	*p* < 0.001

**Table 7 jpm-12-01106-t007:** Factors associated with worse HAM-A test results in terms of Odds Ratio and its 95% C.I. Variables are initially studied univariately and then multivariately by choosing the model with the lowest BIC.

HAM-A Test	Univariate Analysis		Multivariate Analysis	
Variables	Odds Ratio	Significance	Odds Ratio	Significance
Age, years	0.81 (0.66–0.99)	*p* = 0.041		
Education Level, years	0.68 (0.59–0.78)	*p* < 0.001		
BMI, kg/m^2^	1.47 (1.36–1.59)	*p* < 0.001		
ALT, IU/L	1.33 (1.25–1.42)	*p* < 0.001		
AST, IU/L	1.44 (1.33–1.56)	*p* < 0.001		
Hb1Ac (%)	14.63 (7.75–18.94)	*p* < 0.001		
Vitamin D, mg/dL	0.66 (0.60–0.73)	*p* < 0.001	0.50 (0.20–0.90)	*p* < 0.001
Homocysteine µmol/L	1.494 (1.37–1.62)	*p* < 0.001	1.60 (1.39–2.10)	*p* = 0.033
Folate, ng/mL	0.16 (0.09–0.27)	*p* < 0.001	0.30 (0.15–0.50)	*p* = 0.039
Vitamin B12, pg/mL	0.91 (0.88–0.93)	*p* < 0.001		
LDL, mg/dL	1.12 (1.09–1.15)	*p* < 0.001		
NAFLD Severity	9.42 (6.19–14.33)	*p* < 0.001	10.12 (8.15–13.40)	*p* = 0.032

**Table 8 jpm-12-01106-t008:** Factors associated with worse NPI test results in terms of Odds Ratio and its 95% C.I. Variables are initially studied univariately and then multivariately by choosing the model with the lowest BIC.

NPI Test	Univariate Analysis		Multivariate Analysis	
Variables	Odds Ratio	Significance	Odds Ratio	Significance
Age, years	0.85 (0.69–1.04)	*p* = 0.12		
Education Level, years	0.64 (0.55–0.74)	*p* < 0.001		
BMI, kg/m^2^	1.53 (1.41–1.66)	*p* < 0.001		
ALT, IU/L	1.36 (1.28–1.45)	*p* < 0.001		
AST, IU/L	1.47 (1.36–1.60)	*p* < 0.001		
Hb1Ac (%)	22.32 (11.01–45.20)	*p* < 0.001	15.20 (10.20–20.23)	*p* < 0.001
Vitamin D, mg/dL	0.58 (0.50–0.66)	*p* < 0.001	0.50 (0.45–0.65)	*p* < 0.001
Homocysteine µmol/L	1.43 (1.32–1.55)	*p* < 0.001		
Folate, ng/mL	0.10 (0.52–0.20)	*p* < 0.001		
Vitamin B12, pg/mL	0.90 (0.88–0.92)	*p* < 0.001		
LDL, mg/dL	1.12 (1.09–1.15)	*p* < 0.001		
NAFLD Severity	9.01 (5.96–13.61)	*p* < 0.001	8.20 (4.56–14.10)	*p* < 0.001

## Data Availability

Data sharing not applicable.

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
