# Peer review of "Nonalcoholic Fatty Liver Disease and Altered Neuropsychological Functions in Patients with Subcortical Vascular Dementia"

_jpm, 2022, doi:10.3390/jpm12071106_

Round 1

Reviewer 1 Report

This is a very intriguing and important article demonstrating a strong correlation between non-alcoholic fatty liver disease (NAFLD) and small-vessel disease dementia (sVAD) clinical manifestations. Also, a robust relationship between NAFLD and severe vitamin B12, folate, vitamin D hypovitaminosis, and higher hyperhomocysteinemia levels was demonstrated, which may additionally contribute to vascular damage to the brain. 

There are several methodological aspects I need the authors to clarify. Also, I believe the Discussion segment of the paper needs to be revised significantly.

  1. Why do you choose to use the abbreviation sVAD for “small vessel disease-related dementia) instead of e.g. SVD-D or SVD-RD? Is it corresponding to “sVaD” as vascular dementia is also often abbreviated to VaD?
  2. Please provide references in lines 107-109. Why did you select these cut-off values for the folate, vitamin B12, and homocysteine? 
  3. How did you exclude patients with normal pressure hydrocephalus, considering that vascular dementia patients and normal pressure hydrocephalus patients share a lot of clinical characteristics and neuroimaging features? Was this distinction based on clinical, cerebrospinal fluid (CSF), or neuroimaging data, and what criteria did you use? Did these patients undergo lumbar puncture with CSF pressure measurement?
  4. Were examiners (e.g. ultrasonographers) but also other researchers blinded for other laboratory and clinical data as well as for diagnosis of sVAD? How was blinding done?
  5. Since this is a cross-sectional study, the authors can not make a conclusion about the causal relationship between non-alcoholic fatty liver disease and neuropsychological functions in patients with small vessel disease-related dementia but rather only an association. Therefore, the paper title should also be changed accordingly. Also, the authors can not conclude that “our study testifies that NAFLD worsens the clinical presentation of vascular dementia” but rather that there is an association between NAFLD and worse clinical presentation in patients meeting the criteria for vascular dementia (lines 391, 392).
  6. In regard to Table 1, the data given in the table should not be repeated in the text of the Results part of the manuscript (lines 183-194).
  7. In regard to Table 2, the results given in the table should not be repeated in the text of the manuscript (lines 198-204).
  8. Also, in regard to Table 3, the results given in the table should not be stated again in the text of the manuscript (lines 209-220).
  9. In regard to all tables, abbreviations used in the tables should be defined under the table.
  10. The first sentence of the Discussion section of the manuscript should be omitted, as data from the Results were just repeated here. The discussion section should start with the most important findings of your study. Limitations and advantages of your study should be discussed at the end of the Discussion section of the manuscript
  11. The last sentence of the first paragraph of the Discussion part (lines 262-263) should be deleted or elaborated on if related to previous work in this field. 
  12. In regard to the discussion of laboratory parameters in the second paragraph of the Discussion, this discussion should be briefly commented on even if confirming previous findings. What are the practical clinical implications of your findings?
  13. The discussion section of the manuscript, in general, should be elaborated with a comparison of the authors' findings with previous work, and a discussion of the importance and clinical utility of their findings. In the current version of the manuscript, most of the Discussion is the repetition of the study results. If data are lacking, this should be discussed as well in the light of future research directions. The discussion part on potential mechanisms explaining their findings is very interesting but should be shortened and more focused on the study main findings. For example, the discussion on the impact of a high-fat diet on brain functioning seems to elaborate and should be shortened and focused on the topic of the paper. 
  14. In statements covered in lines, 361-367 references should be added, even if just repetition of previous ones (72).

Author Response

Dear Sir/Madam,

Thank you for your compelling work;

  1. Following the literature, we have decided to employ the acronym sVAD to establish a common definition, following the literature: Jellinger KA. Pathology and pathogenesis of vascular cognitive impairment - a critical update. Front Aging Neurosci. 2013;5:17. DOI: 10.33897fnagi.2013.00017; Jellinger KA. The enigma of vascular cognitive disorder and vascular dementia. Acta Neuropathol. 2007;113:349–388.; American Psychiatric Association. Major or Mild Vascular Neurocognitive Disorders. Diagnostic and Statistical Manual of Mental Disorders. 5th ed. Washington DC: American Psychiatric Publishing; 2013: 612–615.; Pohjasvaara T, Mantyla R, Ylikoski R, Kaste M, Erkiniunnti T. Comparison of different clinical criteria (DSM.III, ADDTC, ICD10, NINDS-AIREN, DSM-IV) for the diagnosis of vascular dementia. Stroke. 2003;31: 2952–2957. doi:10.1161/01.STR.31. 12.2952. ; Sinha P, Bharath S, Chandra SR. DSM-5 in vascular dementia. Comparison with other diagnostic criteria in a retrospective study. EC Neurol. 2015;2 (3):135–143.; Caruso P, Signori R, Moretti R. Small vessel disease to subcortical dementia: a dynamic model, which interfaces aging, cholinergic dysregulation and the neurovascular unit. Vasc Health Risk Manag. 2019 Aug 7;15:259-281. DOI: 10.2147/VHRM.S190470. PMID: 31496716; PMCID: PMC6689673.; Moretti R, Caruso P. Small Vessel Disease-Related Dementia: An Invalid Neurovascular Coupling? Int J Mol Sci. 2020 Feb 7;21(3):1095. DOI: 10.3390/ijms21031095. PMID: 32046035; PMCID: PMC7036993.; Moretti R, Caruso P, Storti B, Saro R, Kassabian B, Sala A, Giannini A, Gazzin S. Behavior in subcortical vascular dementia with sight pathologies: visual hallucinations as a consequence of precocious gait imbalance and institutionalization. Neurol Sci. 2020 Nov;41(11):3283-3292. DOI: 10.1007/s10072-020-04445-y. Epub 2020 May 13. PMID: 32405881.)

This has been well-accepted in the general published literature to distinguish the subcortical origin from the general expression VAD (vascular dementia); the other acronyms SVD-RD or SVD-D are not entirely accepted.

  1. We have added the literature references of the cut-off values employed in our Laboratory (lines 114-117, highlighted in yellow)
  2. We have clinically excluded NPH: any of our patients who showed signs of unexplained symmetric gait disturbance, acute onset of the frontal-subcortical pattern of cognitive impairment, and urinary urge incontinence, whose MRI scans showed enlarged ventricles whose comorbidities are not sufficient to explain their symptoms. Moreover, we employed the radiological criteria, as explained in the text (lines 81-90, highlighted in yellow)
  3. Ultrasonographers did not participate in the study; so far, they were not aware of the significance of the given result; in the same way, all the blood samples were examined by Hospital Laboratory members as a routine, and anybody was involved in the study.
  4. We agree with your observation. We have changed the title and modified our conclusive statement, as highlighted on lines 368-379
  5. We resumed the results, highlighted in yellow. Line 192-195
  6. We resumed the results, highlighted in yellow, Line 205-208
  7. We resumed the results, highlighted in yellow, Line 217-224
  8. At the end of the description, all the employed abbreviations have been stated above the table.
  9. The discussion beginning has been simplified and resumed in 257-264, highlighted in yellow; the final part has been expanded with the limits of the study (lines 368-379) highlighted in yellow.
  10. We have elaborated it, as highlighted in yellow, Line 375-377
  11. We have rephrased the paragraph and explained our results within two possible scenarios, as highlighted in yellow, Line 266-281
  12. The remaining part of the discussion has been rewritten, lines 281-334, as highlighted in yellow, following your suggestion, and focusing on our results, with a comparison to the reported studies found
  13. It was done and rephrased, as stated in point 12.

Thank you, again

Reviewer 2 Report

Dear Authors, I read your work entitle “” and here I have my recommendations:

Strengths

§  This work has a big sample size.

§  The way that the data were obtained was adequate and fully explained.

§  The discussion section has a very good literature support

Weaknesses

§  The “Introduction” section is pretty small and has the rational behind this study is not so sound for the reader.

§  Even the size of the sample is big the groups of this study are not well organized. It was expected to see clear groups with severe, moderate and mild symptoms. The second group included patients without or with mild NAFLD. Why those patients were integrated in one group? With this the study has a major issue that the “control group is missing”. I would suggest the authors to split clearly the patient’s groups of the study and exclude the 23 patients with mild NAFLD.

§  Another weakness is that almost all factors has statistically significant differences. So with which criterion the groups are categorized to be compared with? I suppose since we have such effect a cohort study would give more sound data or a retrospective study.

Thank you.

Author Response

We deeply appreciate the time and effort Reviewer 2 employed to evaluate our manuscript.
Attached below a point-to-point response to the Reviewer’s concerns on our study weaknesses.

Q1. “The “Introduction” section is pretty small and has the rational behind this study is not so sound for the reader.”
A1. We have revised the introduction accordingly to the reviewers suggestions.
Q2. “Even the size of the sample is big the groups of this study are not well organized. It was expected to see clear groups with severe, moderate and mild symptoms. The second group included patients without or with mild NAFLD. Why those patients were integrated in one group? With this the study has a major issue that the “control group is missing”. I would suggest the authors to split clearly the patient’s groups of the study and exclude the 23 patients with mild NAFLD.”
A2. The reviewer has pointed out something that was not well specified in the previous version of the manuscript. The sub-classification of individuals was made according to NAFLD US evaluation, which is not always related to symptoms or altered LFTs or other laboratory tests. Therefore, we run intergroup differences and then categorized accordingly (as now stated in the revised version).
Q3: “Another weakness is that almost all factors has statistically significant differences. So with which criterion the groups are categorized to be compared with? I suppose since we have such effect a cohort study would give more sound data or a retrospective study.”
A3: Thank you for the valuable comment. Differences in terms of laboratory values were found and reflects the current literature. However, it is worth stating that the cohort effect is a limiting factor, and more studied are needed (as now stated in the discussion section). This limitation cannot be overcome with the current available data, but this study is the first of its kind. We are confident that
future studies will help.

Round 2

Reviewer 1 Report

Dear Authors,

Your paper has been improved substantially after the revision. However, there are still several issues that need to be addressed:

1.     Discussion, first paragraph, lines 270-272: First two sentences need to be corrected in order to be more informative. I suggest the authors state in the first sentence their study's main results, e.g. “In this cross-sectional study including/comprising 285 sVAD patients, patients with moderate-to-severe NAFLD (Group A) had on average worse (or more disturbed, etc.)  metabolic profile compared to patients with none-to-mild NAFLD (Group B).”

2.     Discussion, second paragraph, third sentence, lines 280-283: Wouldn’t you rather state that “”NAFLD worsens is associated with metabolic aspects and correlates with causes a lack of vitamin B12, folate,…”? Since this is a cross-sectional study, I don’t believe a causal relationship between these parameters has been established, but rather a correlation or an association. In addition, lines 283-288, the fourth sentence in the same paragraph, starting with “The observed worsening could be explained by ….”, to what “worsening” are you specifically referring? The last sentence in the same paragraph (line 288) is redundant: of course, you will discuss your hypotheses in the light of your results.

3.     Discussion, fourth paragraph, lines 300-301: please specify what “conditions have been strictly associated with sVaD worsening” and re-assess if the term “strictly” is appropriate here. Do you believe that evidence linking different vitamin levels and sVaD progression is that strong or is more indicating an association?

4.     Overall, although very comprehensive and interesting, the Discussion section of the paper should be shortened and more focused on the topic, paragraph 7 in particular (lines 327-347).

5.     Statement in lines 358-360 is missing reference.

6.     In sentence in lines 361-363, what did you mean by “diagnosing dementia without any other mention”?

7.     Line 365, what do you mean by “demented brain”, could you please rephrase this statement?

8.     Sentences in lines 385-389 should be merged.

9.     English editing of the manuscript is needed. It is necessary to invest some additional efforts to improve the style and language used in the manuscript. For example, it would be nice to avoid repetition in the Discussion section of the paper (our results, these results…), or to find an adequate substitute for the word “fellows” (line 291). There is inconsistent use of capital letters in certain words (e.g. Homocysteine, line 282), as well as inconsistent use of sVAD and sVaD abbreviations. In addition, please double-check other abbreviations, for example, the small vessel disease (SVD) abbreviation has been introduced but is not consistently used throughout the manuscript (line 303 e.g.). I would also kindly ask the authors to double-check their use of the English language, as there are still some omissions (e.g. line 305, “lead” instead of “leads”, etc.). Please also check for grammar, typing errors, and repetition (lines 354, 356, 359). 

Author Response

Dear Reviewer,

Thank you once again for your efforts and for the contribution you gave to our manuscript.

  1. Your statement has been accomplished and the paragraph rephrased, as highlighted in blue.
  2. We have rephrased the paragraph line 256-267, as highlighted in blue.
  3. We have specified and rephrased the original sentence, as highlighted in blue. We employed a more generic relationship/association factor between the lack of vitamins and the general clinical worsening.
  4. The discussion has been focused on, and some parts are rewritten, highlighted in blue.
  5. The previous 358-360 missing references have been added, but now they can be found ref. 102
  6. We have explained the and rephrased the paragraphs
  7. Rephrased
  8. Done
  9. Done

Reviewer 2 Report

Dear Authors,

I have no further comments on your work. Since all my concerns were addressed.

Thank you.

Author Response

We are thankful to Reviewer 2 valuable comments and for the fact that they have been previously addressed.